# MDSCMF: Matrix Decomposition and Similarity-Constrained Matrix Factorization for miRNA–Disease Association Prediction

**DOI:** 10.3390/genes13061021

**Published:** 2022-06-06

**Authors:** Jiancheng Ni, Lei Li, Yutian Wang, Cunmei Ji, Chunhou Zheng

**Affiliations:** 1Network Information Center, Qufu Normal University, Qufu 273165, China; nijch@163.com; 2School of Cyber Science and Engineering, Qufu Normal University, Qufu 273165, China; wytfuture@163.com (Y.W.); cunmeiji@126.com (C.J.); 3School of Artifial Intelligence, Anhui University, Hefei 230601, China

**Keywords:** miRNA, disease, miRNA–disease association, matrix decomposition, similarity-constrained matrix factorization

## Abstract

MicroRNAs (miRNAs) are small non-coding RNAs that are related to a number of complicated biological processes, and numerous studies have demonstrated that miRNAs are closely associated with many human diseases. In this study, we present a matrix decomposition and similarity-constrained matrix factorization (MDSCMF) to predict potential miRNA–disease associations. First of all, we utilized a matrix decomposition (MD) algorithm to get rid of outliers from the miRNA–disease association matrix. Then, miRNA similarity was determined by utilizing similarity kernel fusion (SKF) to integrate miRNA function similarity and Gaussian interaction profile (GIP) kernel similarity, and disease similarity was determined by utilizing SKF to integrate disease semantic similarity and GIP kernel similarity. Furthermore, we added *L*_2_ regularization terms and similarity constraint terms to non-negative matrix factorization to form a similarity-constrained matrix factorization (SCMF) algorithm, which was applied to make prediction. MDSCMF achieved AUC values of 0.9488, 0.9540, and 0.8672 based on fivefold cross-validation (5-CV), global leave-one-out cross-validation (global LOOCV), and local leave-one-out cross-validation (local LOOCV), respectively. Case studies on three common human diseases were also implemented to demonstrate the prediction ability of MDSCMF. All experimental results confirmed that MDSCMF was effective in predicting underlying associations between miRNAs and diseases.

## 1. Introduction

MiRNAs are 17–24 nt non-coding RNAs that play a pivotal role in controlling the expression of genes through RNA cleavage or translation repression [1,2,3]. Lin-4 was the first miRNA inspected experimentally, by Lee et al. [4] in 1993. Since that time, a large number of miRNAs have been discovered experimentally by researchers [4,5,6]. Researchers have found that various miRNAs are bound up with several crucial biological processes, such as cell development, cell differentiation, cell proliferation, etc. [7,8,9,10]. Developmental defects can be the result of the dysregulation of miRNAs that are associated with the progression of diseases [11]. In the meantime, numerous studies have indicated that miRNAs are connected with a serious of human neoplasms, including breast neoplasms, lung neoplasms, prostate neoplasms, etc. [12,13,14]. Hence, distinguishing miRNAs associated with diseases can deepen the understanding of the genetic causes of complex diseases. Strong connections between miRNAs and diseases have been found by a variety of traditional experiments in the past few years [15,16]. Traditional manual models can infer the associations between miRNAs and diseases, but these are time-consuming, laborious, and have a high failure rate. Therefore, showing the potential relationships between miRNAs and diseases requires effective and stable computational methods, which can obtain increasingly reliable miRNA–disease associations.

In the past, a great number of heterogeneous-network-based algorithms and methods have been applied to predict potential miRNA–disease relationships [17,18,19]. Under the assumption that miRNAs with similar functions have a high probability of being related to diseases with similar phenotypes, and vice versa [20], Jiang et al. [21] established a new calculation-based model that identified potential miRNA–disease connections by applying hypergeometric distribution. However, the similarity information utilized in this model excluded similarity scores. Li et al. [22] constructed a new model that could be used to prioritize human cancer miRNAs by measuring the associations between cancer and miRNAs based on the functional consistency scores of the miRNA target genes and the cancer-related genes. To infer the miRNA–protein connections and disease–protein connections, Mørk et al. [23] built the miPRD model. This model used selective connections to predict the relationships between miRNAs and diseases. Chen et al. [24] utilized the within and between scores of each miRNA–disease combination in the WBSMDA model to predict underlying miRNAs related to diseases. The WBSMDA model also predicted the possible relationships between new diseases and new miRNAs. Yu et al. [25] proposed an identifiable model to infer potential miRNA–disease relationships. This model combined miRNA functional similarity, disease semantic similarity and disease phenotypic similarity to create a modified information flow method. In a phenome–microRNAome network, possible connections and validated relationships between miRNAs and diseases were adopted. Chen et al. [26] introduced the Jaccard similarity between miRNAs and diseases into the BLHARMDA model to investigate prospective miRNA–disease relationships. For improving the prediction efficiency, BLHARMDA used a bipartite local model with a KNN architecture. Ha et al. [27] proposed a computational framework of metric learning named MLMD for predicting potential miRNA–disease associations. MLMD exploited distance metric learning on a miRNA–disease bipartite graph to infer unconfirmed miRNA–disease associations. The excellent performance of MLMD could be attributed to two factors: On the one hand, the implementation of metric learning overcame the violation of triangle inequality. On the other hand, the miRNA expression data were adequately trained in metric learning. Li et al. [28] proposed a similarity-constrained matrix factorization method to infer unconfirmed disease-associated miRNAs. To construct an information-rich similarity matrix, they utilized similarity network fusion to integrate various kinds of similarities. Then, similarity-based regularization terms were added to common non-negative matrix factorization to form a similarity-constrained matrix factorization algorithm, which was applied to make accuracy predictions. The above methods are mainly based on the construction of heterogeneous networks to identify and speculate on the potential disease-related miRNAs, and after cross-validation and case analysis experiments, it was proven that they can be used to observe the potential association between miRNA and disease, but their prediction performance still needs to be improved.

Recently, methods based on the random walk method have gradually been proposed, and more accurate prediction results have been obtained. Shi et al. [29] utilized the function links between human disease genes and miRNA targets to devise a novel model. A random walk algorithm and global network distance measurement were applied to search for feasible miRNA–disease relationships. Chen et al. [30] utilized a random walk with restart algorithm to construct the RWRMDA model. Because the prediction performance calculated by global network similarity was better than the of the local network [31], RWRMDA employed global network similarity to determine the feasible interactions between microRNAs and diseases. Unfortunately, RWRMDA was inappropriate for the diseases without known associated miRNAs. Liu et al. [32] also implemented a random walk with restart algorithm in their model to make prediction results to a higher degree. They employed the random walk with restart algorithm on a heterogeneous graph established by utilizing disease similarity and miRNA similarity. Luo et al. [33] employed an imbalanced bi-random walk method on a heterogeneous network with information on miRNAs and diseases to identify feasible miRNA–disease interactions. When the random walk algorithm is used for association prediction, the initial state of disease nodes and miRNA nodes in the network is very important. Researchers have proposed many design methods for the initial state of nodes in recent years, but the prediction performance has not been greatly improved.

As artificial intelligence technology has developed, machine-learning-based models have increasingly been employed for the accurate prediction of miRNA–disease relationships. To obtain accurate results in matrix completion for miRNA–disease association prediction, Li et al. [34] avoided using negative samples in MCMDA. To infer unknown miRNA–disease interactions, the probabilistic matrix factorization (PMF) algorithm was applied [35] to make predictions. The PMF algorithm is a machine learning technique commonly employed in recommender systems, and can effectively utilize all available data to recommend miRNAs linked to the disease in question. Ha et al. [36] utilized a matrix completion with network regularization method to recognize potential disease-related miRNAs. They considered an miRNA network as additional implicit feedback, and made predictions for disease associations with a given miRNA relying on its direct neighbors. Guo et al. [37] introduced MLPMDA—a novel model for predicting miRNA–disease associations using multilayer linear projection. They processed miRNA–disease interaction information by processing the top nearest neighbors of entities, and then used the updated miRNA–disease interactions and disease similarity to construct a heterogeneous matrix. In this heterogeneous matrix, the multilayer projection and layer-stacking strategy were employed to make predictions. However, in order to obtain dependable and steady performance, MLPMDA requires high-quality biological data. Ding et al. [38] presented a novel computational model named VGAMF for predicting miRNA–disease associations. VGAMF first integrated several different types of information about miRNAs and diseases into comprehensive similarity networks of miRNAs and diseases, which were used to extract the nonlinear representations of miRNAs and diseases based on the variational graph autoencoders. Then, VGAMF obtained the linear representations of miRNAs and diseases by implementing non-negative matrix factorization to process the miRNA–disease association matrix. Finally, a fully connected neural network combined linear representations with nonlinear representations to generate the predicted miRNA–disease association scores. Wang et al. [39] presented a novel method called NMCMDA to observe unknown disease-related miRNAs. The encoder and decoder were the two essential components in NMCMDA. The encoder was developed using a graph neural network to extract latent miRNA and disease characteristics from a heterogeneous miRNA–disease network. These latent features were used by the decoder to generate miRNA–disease association scores. For NMCMDA, a variety of encoders and decoders have been proposed. Finally, in NMCMDA, the combination of a relational graph convolutional network encoder and a neural multirelational decoder achieved the best prediction results. In summary, machine-learning-based models can produce more accurate prediction results, but most of them have difficulties in adjusting the optimal parameters and selecting negative samples, which seriously affect the training efficiency of the model. 

Despite their outstandingly good performance, the abovementioned prediction models have several limitations, such as inadequate measurement of similarity, excessive noise in experimental data, and inaccurate prediction results. To overcome these limitations, we present a novel model called MDSCMF, which combines matrix decomposition with similarity-constrained matrix factorization to predict unobserved miRNA–disease associations. To construct information-rich miRNA similarity and disease similarity, we applied SKF to integrate various kinds of miRNA similarity data and disease similarity data. In addition, because the unknown miRNA–disease associations were much more numerous than the known associations, an MD algorithm was used to get rid of outliers from the miRNA–disease association matrix. Furthermore, we added $L_{2}$ regularization terms and similarity constraint terms to non-negative matrix factorization to form an SCMF algorithm, which was implemented to obtain the final association scores of each miRNA–disease pair. To evaluate the effectiveness of MDSCMF, 5-CV, global LOOCV, and local LOOCV were carried out on the known miRNA–disease association data downloaded from HMDD v3.2 [40]. Furthermore, we performed case studies on colon neoplasms, breast neoplasms, and lung neoplasms for prediction. As a result, 29, 29, and 28 out of the top 30 miRNAs potentially connected to these high-risk human diseases, respectively, were confirmed by miR2Disease [41] and dbDEMC v2.0 [42]. Experimental results showed that MDSCMF was effective for inferring possible relationships between miRNAs and diseases.

## 2. Results

### 2.1. Performance Evaluation

In this section, based on the verified associations between miRNAs and diseases in the HMDD v3.2 database, 5-CV, global LOOCV, and local LOOCV were implemented to evaluate the prediction performance of MDSCMF.

In the framework of 5-CV, we compared MDSCMF with other previous computational methods, including GCAEMDA [43], MSCHLMDA [44], NIMCGCN [45], and HFHLMDA [46]. The full set of verified miRNA–disease associations were divided into five parts in a random manner, where the test set was held by each part in turn, while the training set consisted of the other four parts. The full set of unknown miRNA–disease associations were considered as candidate samples. We applied our method to determine the ranking of the test set relative to candidate samples. Furthermore, for the purpose of reducing potential deviations resulting in random sample segmentations, we applied 100 repeated segmentations to verify the miRNA–disease associations. When the ranking of all test samples was higher than a certain threshold, MDSCMF was regarded as a valid method. Then we could utilize the receiver operating characteristic (ROC) curve that was obtained by plotting the true positive rate (TPR) against the false positive rate (FPR) to effectively evaluate the performance of MDSCMF. We could calculate the areas under the ROC curve (AUCs) of these methods, whose values were between 0 and 1. Figure 1 indicates that MDSCMF, GCAEMDA, MSCHLMDA, NIMCGCN, and HFHLMDA had AUC values of 0.9488, 0.9415, 0.9324, 0.9378, and 0.9301, respectively. The AUC value of MDSCMF was clearly higher than that of the other methods.

In the framework of global LOOCV, MDSCMF was also compared with GCAEMDA, MSCHLMDA, NIMCGCN, and HFHLMDA. The test set was held by each verified miRNA–disease association in turn, while the training set was composed of the other verified associations. The full set of unknown miRNA–disease associations were considered as candidate samples. In addition, we applied MDSCMF to obtain all predicted association scores so that the ranking of the test set relative to the candidate samples could be determined. Similar to 5-CV, we also calculated the AUCs of these methods so as to effectively evaluate their performance. From Figure 2, we can see that MDSCMF, GCAEMDA, MSCHLMDA, NIMCGCN, and HFHLMDA had AUC values of 0.9540, 0.9505, 0.9378, 0.9410, and 0.9321, respectively. Hence, the AUC value of MDSCMF was also higher than that of the other methods.

In the framework of local LOOCV, we also compared MDSCMF with other previous models (i.e., RFMDA [47], BNPMDA [48], ABMDA [49] and VGAMF [38]) to objectively evaluate its performance. In this way, we could determine the ability of MDSCMF to predict the associations between miRNAs and diseases without any verified related miRNAs. For random diseases in the HMDD v3.2 database, the confirmed associations between each disease and all miRNAs were considered as the test set, and remaining associations were regarded as the training set. Similar to the previous two cross-validation methods, the AUC value in local LOOCV still served as the evaluation criterion to reflect the ability of these models. The specific results are shown in Figure 3, which shows that the prediction performance of MDSCMF was better than that of the other models.

### 2.2. Parameter Analysis 

In this section, the parameters ϑ and σ were quantitatively analyzed to research their effects on the prediction performance. ϑ and σ were set as the regularization parameters, which were applied to control the overfitting degree and the smoothness of similarity consistency, respectively. We utilized all combinations of two values $\vartheta \in \left\{2^{-3},2^{-2},\ldots ,2^{3}\right\}$ and $\sigma \in \left\{2^{-3},2^{-2},\ldots ,2^{3}\right\}$ to conduct MDSCMF. The AUC values of 5-CV were applied to evaluate the performance of the model under different combinations of parameters. After various tests were conducted, we concluded that the model obtained the best performance when $\vartheta =2^{2}$ and $\sigma =2^{0}$, as shown in Figure 4.

### 2.3. Effects of Matrix Decomposition Analysis 

In this section, we evaluated the effect of the pre-processing MD step for known miRNA–disease association matrix A on the model’s performance. The AUC values of 5-CV were considered as indicators, and the corresponding ROC curves are shown in Figure 5. In MDSCMF, MD considers the sparsity of the miRNA–disease association matrix, thereby improving the prediction ability of the model. Conversely, MDSCMF without MD disregards the sparsity of the original association matrix; thus, the noise data in the matrix may reduce the accuracy of the prediction. As shown in Figure 5, the AUC value of MDSCMF under the 5-CV framework was 0.9488. In contrast, the AUC value of MDSCMF without MD under the 5-CV framework was 0.9291. The results of the comparison distinctly show that MDSCMF with MD has a higher AUC value compared to that without MD.

### 2.4. Case Studies

For the purpose of demonstrating the effectiveness and accuracy of MDSCMF, we applied an evaluation experiment in this study. We implemented several types of human diseases—i.e., colon neoplasms, breast neoplasms, and lung neoplasms—as case studies to validate the performance of our method. Colon neoplasms are malignancies in the field of medicine that have been confirmed to be associated with several miRNAs [50,51]. Breast neoplasms, which have been observed to be associated with several miRNAs in clinical experiments, have a high incidence rate among women [52]. Lung neoplasms are among the most dangerous malignancies, with the fastest increases in morbidity and mortality [53]. A growing body of evidence indicates that these diseases have close relationships with several miRNAs. The miRNAs associated with these diseases were ranked in line with the prediction scores. Moreover, we utilized two databases—miR2Disease [41] and dbDEMC v2.0 [42]—to check miRNAs that had been ranked. 

As a result, 29, 29, and 28 of the top 30 miRNAs inferred by our model were individually confirmed to be associated with colon neoplasms, breast neoplasms, and lung neoplasms, respectively, according to the miR2Disease [41] and dbDEMC v2.0 [42] databases. Table 1, Table 2 and Table 3 show the corresponding prediction results.

## 3. Materials and Methods

In this paper, we utilized the biological information of miRNAs and diseases to propose a novel method called MDSCMF, which fully extends the advantages of matrix decomposition and similarity-constrained matrix factorization to predict possible miRNA–disease associations. The flowchart of MDSCMF is clearly shown in Figure 6.

### 3.1. Human miRNA–Disease Associations

In this study, we took advantage of miRNA–disease association data from the HMDD v3.2 database [40], which contained 12,446 verified associations between 853 miRNAs and 591 diseases. To make calculation more convenient, we constructed an adjacency matrix $A\in R^{nm\times nd}$ to indicate the miRNA–disease relationships. We set nd and nm to stand for the numbers of diseases and miRNAs, respectively. Specifically, the element A(i,j) is equal to 1 when miRNA $m_{i}$ is proved to be connected with disease $d_{j}$, and otherwise it is equal to 0. Therefore, the matrix *A* contains 12,446 entries that are equal to 1.

### 3.2. MiRNA Functional Similarity

The miRNAs with similar functions have a high probability of being related to diseases that are similar, and vice versa [20]. Therefore, we downloaded the miRNA functional similarity data from http://www.cuilab.cn/files/images/cuilab/misim.zip, accessed on 1 June 2022. For ease of calculation, we constructed the matrix $SM_{1}$ to store the data. The element $SM_{1}\left(m_{i},m_{j}\right)$ represents the value of similarity between miRNA $m_{i}$ and miRNA $m_{j}$.

### 3.3. Disease Semantic Similarity

The directed acyclic graph (DAG) based on the MeSH descriptor [54] can be utilized to describe diseases. DAG(D)=(D,T(D),E(D)) represents the DAG of disease *D*. T(D) denotes the nodes in the DAG that include *D* itself and its ancestor nodes. E(D) denotes the edges in the DAG that connect child nodes with parent nodes directly. The formula to calculate the semantic score of disease *D* is defined as follows:(1)$$DV1\left(D\right)=\sum_{d\epsilon T\left(D\right)}D_{D}1\left(d\right),$$
where the formula to calculate the contribution value $D_{D}1\left(d\right)$ of disease *d* is as follows:(2)$$D_{D}1\left(d\right)=\{\begin{array}{c} \;\;\;\;\;\;\;\;\;\;\;\;\;\;\;\;\;1\;\;\;\;\;\;\;\;\;\;\;\;\;\;\;if\;d=D\\ \max\left\{\Delta *D_{D}1\left(d^{\prime }\right)|d^{\prime }\epsilon children\;of\;d\right\}\;if\;d\neq D \end{array},$$
where Δ is the semantic contribution factor, which was equal to 0.5 in our paper, based on previous literature [55].

The formula to obtain the semantic similarity score between disease $d_{i}$ and disease $d_{j}$ is defined as follows:(3)$$SS_{1}\left(d_{i},d_{j}\right)=\frac{\sum_{t\in T\left(d_{i}\right)\cap T\left(d_{j}\right)}\left(D_{d_{i}}1\left(t\right)+D_{d_{j}}1\left(t\right)\right)}{DV1\left(d_{i}\right)+DV1\left(d_{j}\right)}.$$

Furthermore, for the two diseases of the same layer in a DAG, assuming they have different occurrences in DAGs, it does not make sense to define the semantic contributions of the two diseases for this DAG to be consistent. Objectively speaking, the semantic contribution of high-incidence diseases should be less than that of low-incidence diseases. Consequently, to further optimize the similarity information between diseases, another strategy was introduced to calculate disease semantic similarity following this method [56]. Specifically, the formulae to calculate the semantic score of disease *D* and the contribution values of disease *d* are as follows:(4)$$DV2\left(D\right)=\sum_{d\epsilon T\left(D\right)}D_{D}2\left(d\right),$$
(5)$$D_{D}2\left(d\right)=-\log\left(\frac{the\;number\;of\;DAGs\;including\;d\;}{the\;number\;of\;diseases}\right).$$

Then, the formula to obtain the semantic similarity score between $d_{i}$ and disease $d_{j}$ is as follows:(6)$$SS_{2}\left(d_{i},d_{j}\right)=\frac{\sum_{t\in T\left(d_{i}\right)\cap T\left(d_{j}\right)}\left(D_{d_{i}}2\left(t\right)+D_{d_{j}}2\left(t\right)\right)}{DV2\left(d_{i}\right)+DV2\left(d_{j}\right)}.$$

For the purpose of making the results more accurate, we set two kinds of semantic similarity that were equally important. Therefore, if disease $d_{i}$ and $d_{j}$ had semantic similarity, we calculated the average $SD_{1}\left(d_{i},d_{j}\right)$ of $SS_{1}\left(d_{i},d_{j}\right)$ and $SS_{2}\left(d_{i},d_{j}\right)$ by the following formula:(7)$$SD_{1}\left(d_{i},d_{j}\right)=\frac{SS_{1}\left(d_{i},d_{j}\right)+SS_{2}\left(d_{i},d_{j}\right)}{2}.$$

### 3.4. Gaussian Interaction Profile Kernel Similarity

The miRNAs with similar functions have a high probability of being related to similar diseases, and vice versa [20]. Therefore, the Gaussian interaction profile kernel similarity was applied to determine the miRNA similarity and disease similarity [57,58]. We made vector $K\left(d_{i}\right)$ to represent the interaction profile of disease $d_{i}$ in accordance with whether or not $d_{i}$ had a verified association with each miRNA. Similarly, we made vector $K\left(m_{i}\right)$ to represent the interaction profile $m_{i}$ in accordance with whether or not $m_{i}$ had a verified association with each disease. The equation to calculate the GIP kernel similarity of diseases is defined as follows:(8)$$SD_{2}\left(d_{i},d_{j}\right)=\exp\left(-\rho_{d}∥K\left(d_{i}\right)-K\left(d_{j}\right){∥}^{2}\right),$$
where $\rho_{d}$ is applied to control the kernel bandwidth. The $\rho_{d}\;$ is obtained by normalizing the original bandwidth $\rho_{d}^{\prime }$ by the average number of verified associations with miRNAs per disease, as follows:(9)$$\rho_{d}=\rho_{d}^{\prime }/(\frac{1}{nd}\overset{nd}{\underset{i=1}{\sum }}∥K\left(d_{i}\right){∥}^{2}).$$

Similarly, we used the following equations to calculate the GIP kernel similarity of miRNAs:(10)$$SM_{2}\left(m_{i},m_{j}\right)=\exp\left(-\rho_{m}∥K\left(m_{i}\right)-K\left(m_{j}\right){∥}^{2}\right),$$
(11)$$\rho_{m}=\rho_{m}^{\prime }/(\frac{1}{nm}\sum_{i=1}^{nm}∥K\left(m_{i}\right){∥}^{2}).$$

### 3.5. Integrating Similarity for miRNAs and Diseases

In this section, the similarity kernel fusion [59] was implemented to integrate miRNA functional similarity and GIP kernel similarity into ultimate miRNA similarity. The concrete integration process of miRNA similarity matrices can be divided into the following major steps: 

In the first step, two different miRNA similarities are treated as original miRNA similarity kernels, which are defined as $SM_{n}$, n=1, 2 in the above sections. Each miRNA similarity is normalized by the following equation:(12)$$F_{n}\left(m_{i},m_{j}\right)=\frac{SM_{n}\left(m_{i},m_{j}\right)}{\sum_{m_{k}\in M}SM_{n}\left(m_{k},m_{j}\right)},$$
where $F_{n}\left(m_{i},m_{j}\right)$ denotes the normalized kernel that satisfies $\underset{m_{k}\in M}{\sum }F_{n}\left(m_{i},m_{j}\right)=1$, and $M={\left\{m_{i}\right\}}_{i=1}^{nm}$ indicates the set of miRNAs.

In the second step, the neighbor-constraint kernel for each miRNA original kernel can be constructed as follows:(13)$$S_{n}\left(m_{i},m_{j}\right)=\{\begin{array}{c} \frac{SM_{n}\left(m_{i},m_{j}\right)}{\sum_{m_{k}\in N_{i}}SM_{n}\left(m_{i},m_{k}\right)}\;if\;m_{j}\in N_{i}\\ \;\;\;\;\;\;\;\;0\;\;\;\;\;\;\;\;\;if\;m_{j}\notin N_{i} \end{array},$$
where $S_{n}\left(m_{i},m_{j}\right)$ denotes a neighbor-constraint kernel that obeys $\underset{m_{k}\in M}{\sum }S_{n}\left(m_{i},m_{j}\right)=1$, and $N_{i}$ denotes the collection of all neighbors of miRNA $m_{i}$, including itself.

In the third step, the normalized kernels and neighbor-constraint kernels are integrated as follows:(14)$$F_{n}^{l+1}=\tau \left(S_{n}\times \frac{\sum_{t\neq n}F_{t}^{l}}{2}\times S_{n}^{T}\right)+\left(1-\tau \right)\frac{\sum_{t\neq n}F_{t}^{0}}{2},$$
where $F_{n}^{l+1}$ represents the value of n-th kernel after l+1 iterations, $P_{t}^{0}$ represents the initial value of $F_{t}$, and the weight parameter τ∈(0,1) is used to balance the rate. After $F_{n}^{l+1}$, n=1, 2 is obtained, the overall kernel $SM^{*}$ can be calculated by the following formula:(15)$$SM^{*}=\frac{1}{2}\sum_{n=1}^{2}F_{n}^{l+1}.$$

In the fourth step, a weighted matrix W is applied to further eliminate noises in the overall kernel $SM^{*}$. The construction process of W is as follows:(16)$$W\left(m_{i},m_{j}\right)=\{\begin{array}{c} 1\;\;\;\;if\;m_{i}\in N_{j}\cap m_{j}\in N_{i}\\ 0\;\;\;\;if\;m_{i}\notin N_{j}\cap m_{j}\notin N_{i}\\ 0.5\;\;\;\;\;\;otherwise\;\;\;\;\;\; \end{array}.$$

In the last step, the ultimate miRNA similarity kernel $SM\in R^{nm\times nm}$ can be calculated by the following formula:(17)$$SM=W\times SM^{*}.$$

In the same light, we could obtain the ultimate disease similarity kernel as $SD\in R^{nd\times nd}$.

### 3.6. Matrix Decomposition

From the published literature [60], we found that the data used in experiments were far from perfect. Several real data of miRNA–disease associations were redundant and/or missing. Therefore, we decomposed the adjacency matrix A into two sections: The linear combination of the adjacency matrix A and low-rank matrix Y was the first section. The second section was the sparse matrix *X*, which included a large number of zero values. Clearly, the data of the sparse matrix *X* can be regarded as outliers. The matrix decomposition method was applied to acquire the lowest-rank matrix, which was employed to reconstruct a novel adjacency matrix. The formula to decompose the adjacency matrix A is defined as follows:(18)A=AY+X.

For the purpose of making the Y become low-rank, we could enforce nuclear norm on Y. In addition, the $L_{2,1}$ norm was enforced on the *X* so that *X* became sparse. The specific process can be represented by the following formula:(19)$$\underset{Y,X}{\min}∥Y{∥}_{*}+\varphi ∥X{∥}_{2,1}s.t.A=AY+X,$$
where ${∥Y∥}_{*}=\underset{i}{\sum }\beta_{i}\left(i.e.,\;\beta_{i}\;is\;the\;sigular\;values\;of\;Y\right)$ represents the nuclear norm of Y, ${∥X∥}_{2,1}=\overset{n}{\underset{j=1}{\sum }}\sqrt{\overset{n}{\underset{i=1}{\sum }}{\left(X_{ij}\right)}^{2}}$ represents the $L_{2,1}$ norm of X, and the positive free parameter φ is applied to balance the weights of ${∥Y∥}_{*}$ and ${∥X∥}_{2,1}$. Furthermore, minimizing the nuclear norm of Y and the $L_{2,1}$ norm of *X* contributes to convenient calculation.

If the matrix A combined with Y is treated as an identity matrix, the algorithm will become the robust PCA. Therefore, Equation (19) can be treated as a robust PCA generalization [61] and changed into a comparable problem, as follows:(20)$$\underset{Y,X,J}{\min}{∥J∥}_{*}+\varphi {∥X∥}_{2,1}s.t.A=AY+X,\;Y=J.$$

Equation (20) is a constraint and convex optimization problem. Therefore, both the first-order information and the special properties of these convex optimization problems can be employed to solve the issue of scalability. The inexact augmented Lagrange multipliers (IALM) algorithm [62] can be utilized to convert Equation (20) to an unconstraint problem. Then, the augmented Lagrange function is adopted to minimize this problem, as follows:(21)$$L={∥J∥}_{*}+\varphi {∥X∥}_{2,1}+tr\left(F_{1}^{T}\left(A-AY-X\right)\right)+tr\left(F_{2}^{T}\left(Y-J\right)\right)+\frac{\alpha }{2}\left(∥A-AY-X{∥}_{F}^{2}+∥Y-J{∥}_{F}^{2}\right),$$
where the penalty parameter α≥0. After minimization with respect to J, Y, and X, the above problem can be settled effectively. In addition, Equation (21) can be solved when the other variables are fixed and the Lagrange multipliers $F_{1}$ and $F_{2}$ are updated. The specific steps for solving Equation (21) are displayed in Algorithm 1.
**Algorithm 1:** Solving Equation (21) by IALM**Input:** Given an incomplete matrix A and parameters φ∈{0,1}

**Output:**
Y* and X***Initialize:**Y=0, X=0, F1=0, F2=0, α=10−4, maxα=1010, γ=1.1, ε=10−8**while** not converged do

1: Fix the other and update J by $J=arg\;min\frac{1}{\alpha }{∥J∥}_{*}+∥\frac{1}{2}J-\left(Y+\frac{F_{2}}{\alpha }\right){∥}_{F}^{2}$

2: Fix the other and update Y by $Y={\left(I+A^{T}A\right)}^{-1}$($A^{T}A-A^{T}X+J+\left(A^{T}F_{1}-F_{2}\right)/\alpha$)

3: Fix the other and update X by $X=\arg min\frac{\varphi }{\alpha }{\left|\left|X\right|\right|}_{2,1}+∥\frac{1}{2}X-\left(AY+F_{1}/\alpha \right){∥}_{F}^{2}$

4: Update the multiplier $F_{1}=F_{1}+\alpha \left(A-AY-X\right);\;F_{2}=F_{2}+\alpha \left(Y-J\right)$

5: Update parameter α by $\alpha =\min\left(\gamma \alpha ,max_{\alpha }\right)$

6: Check the convergence condition ${\left|\left|A-AY-X\right|\right|}_{\infty }<\varepsilon \;and\;{\left|\left|Y-J\right|\right|}_{\infty }<\varepsilon \;$

**end while**

We defined the solution of Equation (21) as $Y^{*}$ and $X^{*}.$ The A(i,j) was used to represent the association between miRNA $m_{i}$ and disease $d_{j}$, so $Y^{*}\in R^{nd\times nd}$ could be applied to represent the similarity between diseases. When $Y^{*}$ was obtained, the adjacency matrix $A^{*}$ denoted new associations between miRNAs and diseases that could be calculated by the following equation:(22)$$A^{*}=AY^{*}.$$

### 3.7. Similarity-Constrained Matrix Factorization

In this section, the $L_{2}$ regularization terms and similarity constraint terms were added to a traditional non-negative matrix factorization algorithm to form similarity-constrained matrix factorization, which was applied to observe more unknown miRNA–disease interactions. The matrix $A^{*}\in R^{nm\times nd}$ can be factorized into $U\in R^{nm\times \gamma }$ and $V\in R^{nd\times \gamma }$, where γ represents the dimensions of miRNA features and disease features. Concretely, the miRNA–disease association can be regarded as the inner product between the miRNA feature vector and the disease feature vector: $a^{*}{}_{ij}\approx u_{i}v_{j}^{T}$, where $a^{*}{}_{ij}$ indicates the (i,j)th element of matrix $A^{*}$, while $u_{i}$ and $v_{j}$ indicate the ith row of U and the jth row of V, respectively. The corresponding objective function is defined as follows:(23)$$min\frac{1}{2}\underset{ij}{\sum }{\left(a^{*}{}_{ij}-u_{i}v_{j}^{T}\right)}^{2}.$$

In what follows, the $L_{2}$ regularization terms of $u_{i}$ and $v_{j}$ are added to above function for preventing overfitting in the model:(24)$$min\frac{1}{2}\underset{ij}{\sum }{\left(a^{*}{}_{ij}-u_{i}v_{j}^{T}\right)}^{2}+\frac{\vartheta }{2}\underset{i}{\sum }∥u_{i}{∥}^{2}+\frac{\vartheta }{2}\underset{j}{\sum }∥v_{j}{∥}^{2},$$
where ϑ denotes the regularization parameter for controlling the balance. 

When data points are mapped from high-rank space into low-rank space, the geometric properties of the data points will most likely stay the same [63,64]. Owing to the miRNA similarity SM and disease similarity SD being able to represent the geometric structure of the data points, the similarity constraint terms $S_{U}$ and $S_{V}$ are proposed as follows:(25)$$S_{U}=\frac{1}{2}\underset{ij}{\sum }∥u_{i}-u_{j}{∥}^{2}SM_{ij},$$
(26)$$S_{V}=\frac{1}{2}\underset{ij}{\sum }∥v_{i}-v_{j}{∥}^{2}SD_{ij},$$
where $SM_{ij}$ represents the similarity between miRNAs $m_{i}$ and $m_{j}$, while $SD_{ij}$ denotes the similarity between diseases $d_{i}$ and $d_{j}$. Because the degree of similarity between two random data points is determined by the distance between them, $S_{U}$ will incur a heavy penalty if the distance between $m_{i}$ and $m_{j}$ is close in the miRNA feature space. Thus, we minimized the $S_{U}$ to keep the geometric structure of the miRNA data points, which would give rise to $m_{i}$ and $m_{j}$ being mapped closely in low-dimensional space. The same is true for disease data nodes, so we also minimized the $S_{V}$. Based on the above analysis, the objective function of SCMF can be defined by adding $S_{U}$ and $S_{V}$ to Equation (24) as follows:(27)$$\underset{U,V}{\min}L=\frac{1}{2}\underset{ij}{\sum }{\left(a^{*}{}_{ij}-u_{i}v_{j}^{T}\right)}^{2}+\frac{\vartheta }{2}\underset{i}{\sum }∥u_{i}{∥}^{2}+\frac{\vartheta }{2}\underset{j}{\sum }∥v_{j}{∥}^{2}+\frac{\sigma }{2}\underset{ij}{\sum }∥u_{i}-u_{j}{∥}^{2}SM_{ij}+\frac{\sigma }{2}\underset{ij}{\sum }∥v_{i}-v_{j}{∥}^{2}SD_{ij},$$
where σ denotes the hyperparameter to control the smoothness degree of similarity consistency. Subsequently, an efficacious optimization algorithm is proposed to calculate the objective function of SCMF. 

First, the partial derivatives of L with respect to $u_{i}$ and $v_{j}$ can be calculated by the following formulae:(28)$$\;\;\;\;\;\;\;\;\;\;\nabla_{u_{i}}L=\underset{j}{\sum }\left(u_{i}v_{j}^{T}-a^{*}{}_{ij}\right)v_{j}+\vartheta u_{i}+\sigma \left(\underset{j}{\sum }\left(u_{i}-u_{j}\right)SM_{ij}-\underset{j}{\sum }\left(u_{j}-u_{i}\right)SM_{ji}\right)\;\;\;=u_{i}\left(V^{T}V+\vartheta I+\sigma \left(\underset{j}{\sum }SM_{ij}+\underset{j}{\sum }SM_{ji}\right)I\right)-A^{*}\left(i,:\right)V-\sigma \underset{j}{\sum }\left(SM_{ij}+SM_{ji}\right)u_{j},$$
(29)$$\;\;\;\;\;\;\;\;\;\;\nabla_{v_{j}}L=\underset{j}{\sum }\left(v_{j}u_{i}^{T}-a^{*}{}_{ij}\right)u_{i}+\vartheta v_{j}+\sigma \left(\underset{i}{\sum }\left(v_{j}-v_{i}\right)SD_{ji}-\underset{i}{\sum }\left(v_{i}-v_{j}\right)SD_{ij}\right)\;\;\;=v_{j}\left(U^{T}U+\vartheta I+\sigma \left(\underset{i}{\sum }SD_{ij}+\underset{i}{\sum }SD_{ji}\right)I\right)-A^{*}{\left(:,j\right)}^{T}U-\sigma \underset{i}{\sum }\left(SD_{ij}+SD_{ji}\right)v_{i},$$
where $A^{*}\left(i,:\right)$ and $A^{*}\left(:,j\right)$ indicate the  ith row and jth column of matrix $A^{*}$, respectively.

Next, the calculation of the second derivatives of L with respect to $u_{i}$ and $v_{j}$ is presented as follows:(30)$$\nabla_{u_{i}}^{2}L=V^{T}V+\vartheta I+\sigma \left(\underset{j}{\sum }SM_{ij}+\underset{j}{\sum }SM_{ji}\right)I,$$
(31)$$\nabla_{v_{j}}^{2}L=U^{T}U+\vartheta I+\sigma \left(\underset{i}{\sum }SD_{ij}+\underset{i}{\sum }SD_{ji}\right)I.$$

Then, $u_{i}$ and $v_{j}$ can be iteratively updated according to Newton’s method, as follows:(32)$$u_{i}\leftarrow u_{i}-\nabla_{u_{i}}L{\left(\nabla_{u_{i}}^{2}L\right)}^{-1},$$
(33)$$v_{j}\leftarrow v_{j}-\nabla_{v_{j}}L{\left(\nabla_{v_{j}}^{2}L\right)}^{-1}.$$

More specifically, the update of $u_{i}$ and $v_{j}$ can be performed by the below formulas:(34)$$u_{i}\leftarrow \left(A^{*}\left(i,:\right)V+\sigma \underset{j}{\sum }\left(SM_{ij}+SM_{ji}\right)u_{j}\right){\left(V^{T}V+\vartheta I+\sigma \left(\underset{j}{\sum }SM_{ij}+\underset{j}{\sum }SM_{ji}\right)I\right)}^{-1},$$
(35)$$v_{j}\leftarrow \left(A^{*}{\left(:,j\right)}^{T}U+\sigma \underset{i}{\sum }\left(SD_{ij}+SD_{ji}\right)v_{i}\right){\left(U^{T}U+\vartheta I+\sigma \left(\underset{i}{\sum }SD_{ij}+\underset{i}{\sum }SD_{ji}\right)I\right)}^{-1}.$$

The update of $u_{i}$ and $v_{j}$ will stop when the convergence condition is satisfied. After that, the predicted association matrix can be calculated by the following formula:(36)$$A^{\prime }=UV^{T}.$$

The value of $A_{ij}^{\prime }$ denotes the predicted association score between miRNA $m_{i}$ and disease $d_{j}$. The higher the prediction score, the greater the association probability.

## 4. Discussion

To solve the problems of inadequate measurement of similarity, excessive noise in experimental data, and inaccurate prediction results existing in previous prediction models, we developed a computational model for predicting miRNA–disease associations based on matrix decomposition and similarity-constrained matrix factorization (MDSCMF). Because the miRNA–disease association matrix was a sparse matrix, we applied the MD algorithm to complete it. Our results demonstrated that the MD algorithm could improve the prediction performance to some extent. In addition, we applied SKF to integrate various types of similarities for constructing information-rich miRNA similarity and disease similarity. Furthermore, $L_{2}$ regularization terms and similarity constraint terms were added to non-negative matrix factorization to form the SCMF algorithm, which was utilized to generate association scores of each miRNA–disease pair. In the frameworks of 5-CV, global LOOCV, and local LOOCV, the AUCs of MDSCMF achieved 0.9488, 0.9540, and 0.8672, respectively, indicating that the performance of our method had a significant improvement relative to previous methods. Furthermore, the predicted miRNAs related to colon neoplasms, prostate neoplasms, and lung neoplasms were confirmed by the experimental literature, so the prediction results generated by our method were proven to be reliable.

It should be noted that the following factors may contribute to the reliable performance of MDSCMF: First of all, the MD algorithm, which greatly alleviated the influence of the inherent noise existing in the current dataset, was utilized to refine the miRNA–disease association matrix. In addition, when we used SCMF to make predictions, the $L_{2}$ regularization terms and similarity constraint terms could avoid overfitting problems and generate robustness of the data richness, respectively.

However, several limitations may influence the performance of MDSCMF. First of all, although the amount of data had increased, we still ought to spare no effort to expand the experimental data. Furthermore, the data we utilized included miRNA function similarity data and disease semantic similarity data, which may contain noise and outliers. Therefore, we should continuously optimize our model to improve its performance in the future.

## Figures and Tables

**Figure 1 genes-13-01021-f001:**
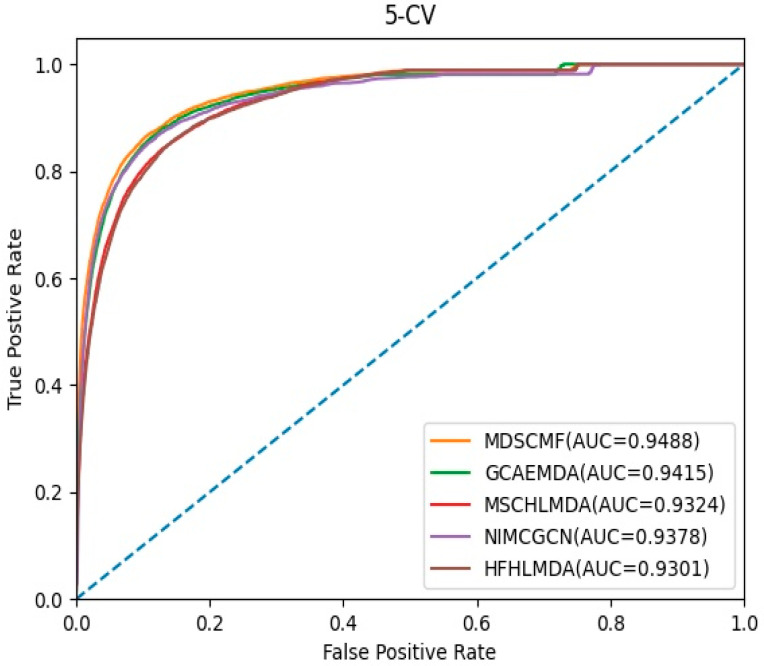
AUC of 5-CV compared with those of GCAEMDA, MSCHLMDA, NIMCGCN, and HFHLMDA.

**Figure 2 genes-13-01021-f002:**
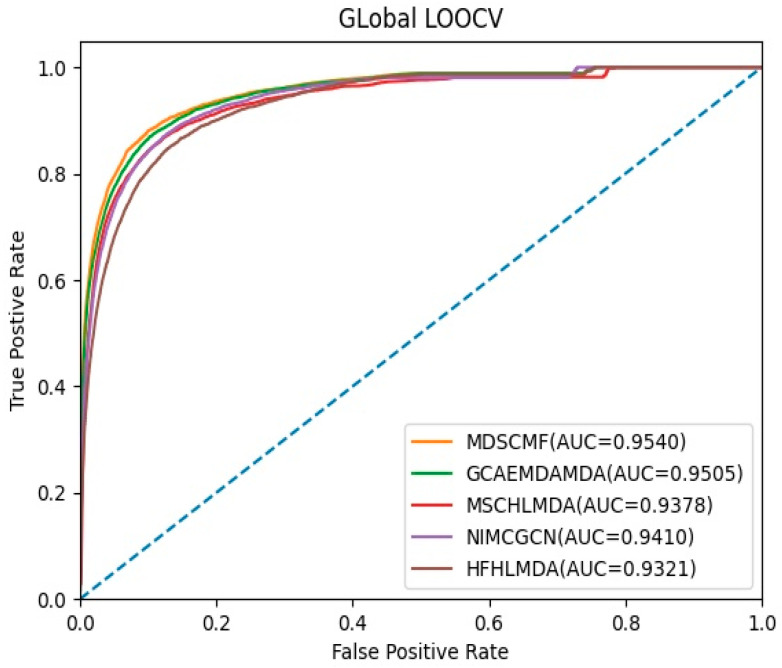
AUC of global LOOCV compared with those of GCAEMDA, MSCHLMDA, NIMCGCN, and HFHLMDA.

**Figure 3 genes-13-01021-f003:**
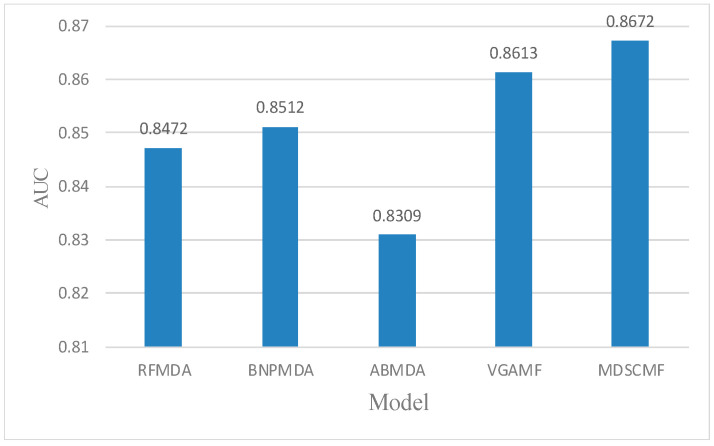
Comparisons between MDSCMF and other computational models by local LOOCV.

**Figure 4 genes-13-01021-f004:**
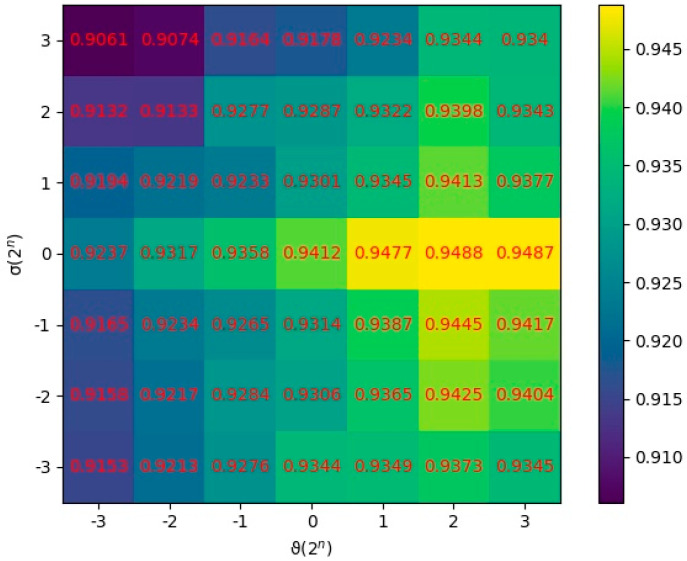
AUCs at different values of ϑ and σ.

**Figure 5 genes-13-01021-f005:**
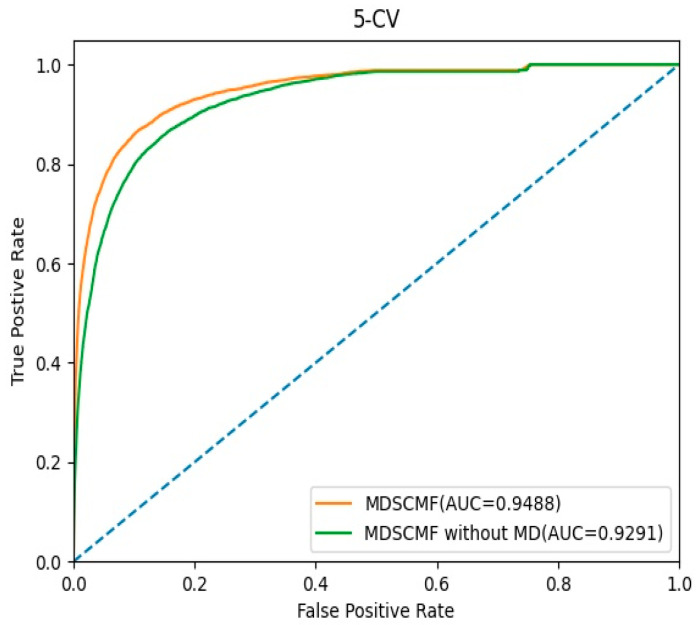
The ROC curves of MDSCMF and MDSCMF without MD.

**Figure 6 genes-13-01021-f006:**
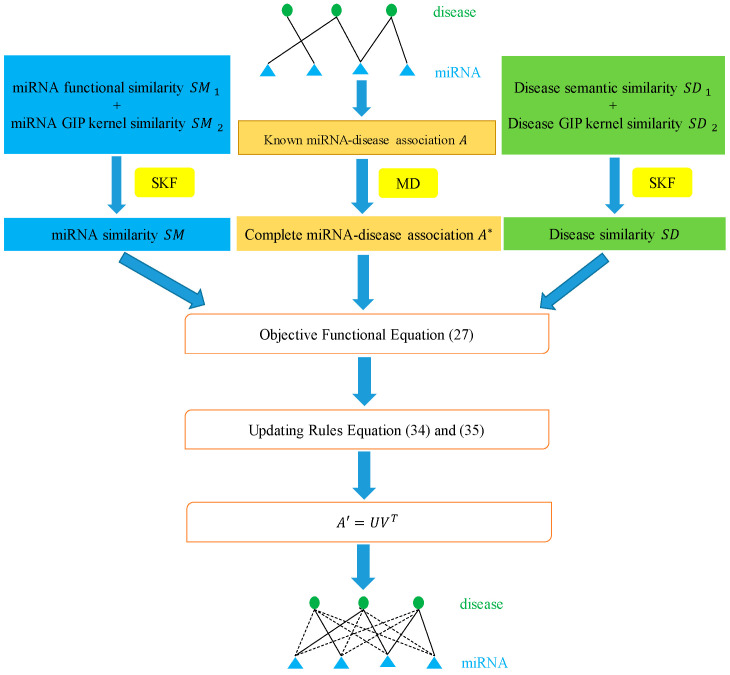
Flowchart of MDSCMF.

**Table 1 genes-13-01021-t001:** The top 30 potential miRNAs associated with colon neoplasms.

miRNA	Evidence	miRNA	Evidence
hsa-mir-630	d	hsa-mir-29b	m; d
hsa-mir-20a	m; d	hsa-mir-141	m; d
hsa-mir-143	m; d	hsa-mir-132	m; d
hsa-mir-584	d	hsa-mir-19b	m; d
hsa-mir-506	d	hsa-mir-29a	m; d
hsa-mir-552	d	hsa-mir-223	d
hsa-mir-128	unconfirmed	hsa-let-125b	d
hsa-mir-7i	m; d	hsa-mir-622	d
hsa-mir-127	m; d	hsa-mir-18a	d
hsa-mir-1290	d	hsa-mir-143	d
hsa-mir-493	d	hsa-mir-125a	m; d
hsa-mir-498	d	hsa-mir-21	m; d
hsa-mir-107	m; d	hsa-mir-137	m; d
hsa-mir-191	m; d	hsa-mir-424	d
hsa-mir-32	m; d	hsa-mir-200b	d

m: miR2Disease database; d: dbDEMC v2.0 database.

**Table 2 genes-13-01021-t002:** The top 30 potential miRNAs associated with breast neoplasms.

miRNA	Evidence	miRNA	Evidence
hsa-mir-99a	m; d	hsa-mir-663	m
hsa-mir-542	d	hsa-mir-520h	d
hsa-mir-96	d	hsa-mir-519d	d
hsa-mir-98	m; d	hsa-mir-186	d
hsa-mir-185	d	hsa-mir-381	d
hsa-mir-130a	d	hsa-mir-32	d
hsa-mir-708	d	hsa-mir-590	unconfirmed
hsa-mir-150	d	hsa-mir-330	d
hsa-mir-192	d	hsa-mir-433	d
hsa-mir-196b	d	hsa-mir-942	d
hsa-mir-888	d	hsa-mir-661	m; d
hsa-mir-9	m; d	hsa-mir-337	d
hsa-mir-130b	d	hsa-mir-494	d
hsa-mir-592	d	hsa-mir-212	d
hsa-mir-99b	d	hsa-mir-618	d

m: miR2Disease database; d: dbDEMC v2.0 database.

**Table 3 genes-13-01021-t003:** The top 30 potential miRNAs associated with lung neoplasms.

miRNA	Evidence	miRNA	Evidence
hsa-mir-96	d	hsa-mir-937	unconfirmed
hsa-mir-145	m; d	hsa-mir-30e	m
hsa-mir-99a	m; d	hsa-mir-151	d
hsa-mir-9	m; d	hsa-mir-614	d
hsa-mir-185	d	hsa-mir-1323	d
hsa-mir-130a	d	hsa-mir-32	d
hsa-mir-7	m; d	hsa-mir-1298	d
hsa-mir-150	m; d	hsa-mir-330	d
hsa-mir-192	m; d	hsa-mir-433	d
hsa-mir-769	unconfirmed	hsa-mir-522	d
hsa-mir-939	d	hsa-mir-449a	d
hsa-mir-98	m; d	hsa-mir-143	m; d
hsa-mir-130b	m; d	hsa-mir-564	d
hsa-mir-638	d	hsa-mir-212	m; d
hsa-mir-99b	d	hsa-mir-615	unconfirmed

m: miR2Disease; d: dbDEMC v2.0 database.

## Data Availability

All data are present within the manuscript or available by request to corresponding author, Lei Li (sdwfll1996@163.com).

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
