# Peer review of "MDSCMF: Matrix Decomposition and Similarity-Constrained Matrix Factorization for miRNA–Disease Association Prediction"

_genes, 2022, doi:10.3390/genes13061021_

Round 1

Reviewer 1 Report

Recommendation: Author should prepare a major revision

In this paper, the authors presented a Matrix Decomposition and Similarity Constrained Matrix Factorization (MDSCMF) to predict potential miRNA-disease associations.

1. It is suggested to apply other human miRNA-disease association datasets such as miR2disease and dbDEMC.

2. Literature review is seriously incomplete. Many important computational models and reviews for miRNA-disease association prediction published in the previous journals, such as the paper with pmid: 34252084 and DOI: 10.1109/ACCESS.2021.3084148. 

3. It is suggested to carry out another comparitive experiment. The author should compare MDSCMF with other preivious methods including (pmid: 32260218, DOI: 10.1109/JBHI.2021.3088342) in the framework of local LOOCV.

4. The authors should revise English writing carefully and eliminate small errors in the paper to make the paper easier to understand.

Reviewer 2 Report

The manuscript with the title “MDSCMF: Matrix Decomposition and Similarity Constrained Matrix Factorization for miRNA-Disease Association Prediction” describes the construction of a model for the miRNA-disease association prediction. Even though the application of the model may appear beneficial for the relative research field, important issues can be found in the manuscript as explained below:

  • The Introduction section is extremely long without a synopsis of background knowledge. On the contrary, the Discussion section is extremely short without a critical discussion, integrating the results of the manuscript with current scientific knowledge. The most important issue is that the reader can not properly assess the advantage of this model to other models that exist.
  • It is highly recommended to provide as supplementary material the steps that were performed in this model using simple examples in order to facilitate the repetition of the study and the use of the model for the scientific community.
  • The application of LOOCV and 5-CV seems to not be included in the Materials and Methods section.
  • The phrase “However, several diseases named specific diseases barely participate in a little DAGs, so these diseases ought to gain a higher semantic contribution in DAGs” is not clear.
  • The Materials and Methods section is one section and it is suggested to not separate them.
  • It is highly recommended to improve the language of the manuscript.

Round 2

Reviewer 1 Report

Some comments mentioned in the previous revision were not addressed well.

Still, the author should consider review the previous models, such as(pmid: 32260218, DOI: 10.1109/JBHI.2021.3088342)

Reviewer 2 Report

The manuscript has been improved. Nevertheless, I still suggest including basic methodological information about LOOCV and 5-CV in the Materials and Methods section for the completeness of the manuscript. Additionally, some language expression mistakes still exist.
